# Frontline Involvement in Population COVID-19 Vaccinations: Lived Experience of Nursing Students

**DOI:** 10.3390/healthcare10101985

**Published:** 2022-10-10

**Authors:** Giulia Villa, Emanuele Galli, Sara Allieri, Riccardo Baldrighi, Adelaide Brunetti, Noemi Giannetta, Duilio Fiorenzo Manara

**Affiliations:** 1Center for Nursing Research and Innovation, Faculty of Medicine and Surgery, Vita-Salute San Raffaele University, 20132 Milan, Italy; 2Nursing School San Raffaele Hospital, Vita-Salute San Raffaele University, 20132 Milan, Italy; 3IRCSS San Raffaele Scientific Institute, 20132 Milan, Italy; 4School of Nursing, UniCamillus—Saint Camillus International University of Health and Medical Sciences, 00131 Rome, Italy

**Keywords:** nursing students, vaccination, COVID-19, qualitative research

## Abstract

(1) Background: The globally promoted vaccination campaign has been shown to be the solution for the COVID-19 pandemic, reducing transmission, hospitalisation and the need for intensive care. Although several studies have examined the experiences of healthcare workers during the pandemic, few studies have investigated healthcare student experiences. The aim of this study is to explore the lived experience of third-year nursing students during the COVID-19 vaccination campaign. (2) Methods: A phenomenological qualitative method was adopted. The researchers selected third-year students undertaking a bachelor’s nursing degree who took part in the COVID-19 vaccination campaign at a high-volume vaccination centre in the period from May to August 2021. (3) Results: Thirteen students were included in the study. Three themes and twelve subthemes emerged from the study. The themes were: a challenging experience; it is not as easy as it seems; a learning experience worth living; and teamwork and trust leading to professional development. (4) Conclusions: Participation in the vaccination campaign was a novelty for students in their degree program. Students emphasized the positive aspects of having the opportunity to participate in the vaccination campaign and help the entire community in the fight against COVID-19.

## 1. Introduction

In February 2020, the World Health Organization (WHO) announced that the respiratory disease caused by the new coronavirus is called COVID-19 [1]. The new disease has placed significant pressure on healthcare facilities, testing the resilience and coping skills of healthcare providers, who are at higher risk of stress and burnout [2,3,4,5,6,7,8]. A solution to control the progress of the pandemic is the availability of effective vaccines, which can reduce transmission, hospital admissions, and the need for intensive care [9,10].

Vaccines represent one of the greatest breakthroughs in public health, and vaccination programs have reduced the mortality and morbidity of several infectious diseases, contributing, for example, to the eradication of smallpox, one of the most aggressive infectious diseases in Europe in the 19th century [11].

Although several studies have examined the experiences of healthcare workers during the pandemic, few studies have examined the lived experiences of health profession students undertaking clinical activities. Locally, nursing students have had an important role in the COVID-19 vaccination campaign. According to Goni-Fuste et al., students’ willingness to work alongside nurses during the pandemic depends on a few factors [12]: perceived good health; previous volunteer experience; availability of personal protective equipment (PPE); knowledge about the virus, its modes of transmission, and pandemic management; self-confidence and self-efficacy; perceived support from the university; and a moral duty to help their future colleagues. Previous evidence of nursing students’ experiences and competence in vaccination has pointed out that there are some barriers to effective learning, and an improvement in vaccination-related training is suggested to close the gap [13,14]. To date, the lived experience of nursing students who play an active role in vaccine education and inoculation at vaccination centres has not been investigated.

This study aims to explore the lived experience of third-year students undertaking a bachelor’s nursing degree at the University Hospital in a big city in Northern Italy during the COVID-19 vaccination campaign.

## 2. Materials and Methods

### 2.1. Design

In relation to the aim of the study, a phenomenological qualitative method was adopted. The study was performed according to the Standards for Reporting Qualitative Research and following COREQ guidelines [15,16] (available on Appendix A).

### 2.2. Setting and Participants

For the study, the researchers selected third-year students from a bachelor’s nursing degree who took part in the COVID-19 vaccination campaign at a high-volume vaccination centre (VC) in the period from May to August 2021. They were students in the final year of their course (50 in total). They took part in the vaccination campaign as volunteers. The campaign started at the VC in April 2021, and continued up to the end of July. The centre guaranteed the administration of over 300,000 doses of the vaccine, with a number of daily vaccinations between 1091 and 5031, and a daily average of more or less 3000 vaccinations. The students were selected by two nurses with qualitative research experience (GV, NG) through purposive sampling in order to increase the likelihood of including students able to provide a rich account of their experience [17]. The most “informative” subjects for the purpose of the research were invited to participate; these were students who had just finished (within one week) their experience at the VC. According to the Interpretative Phenomenological Analysis (IPA) methodology, a small homogenous sample was required [18]. Students were recruited until data saturation was reached [19]. Specifically, thirteen students were included in the study.

### 2.3. Data Collection

In accordance with the IPA methodology, a semi-structured interview outline was defined, oriented to understand students’ perceptions and personal experiences during the COVID-19 vaccination campaign. For this reason, open questions were asked which opened up the topic of the experiences had at the vaccination centre, and allowed the participant to give his or her answer in a way that was meaningful to him or her [20]. The questions were developed by the research team with the supervision of a qualitative research expert. Table 1 shows the interview outline.

Data were collected by a researcher (GV) who was not directly involved in the students’ education, and who had expertise in the qualitative research method of semi-structured interviews conducted either in-person or by videocall on an online platform. These were held from June to July 2021. The videocall using an online platform appears to be more convenient, efficient, cost-effective, and flexible than in-person interviews [21]. This mode was inevitably adopted due to the restrictions imposed by the COVID-19 pandemic, allowing interviewees to participate in the study in complete safety. These interviews were audio-recorded and transcribed verbatim. Each participant was asked verbally and in writing for consent to participate in this study and to audio-record the interview. The duration of the interviews was between 15 and 30 min with a total recorded time of 197.16 min.

Data were also collected on age, gender, marital status, number of children, COVID-19 experience (respondents were asked if they had experienced any major COVID-19 sequelae or deaths related to COVID-19 disease), education (the participants were asked if they felt their preparation was sufficient to cope with the experience), information (the students were asked if they felt they received enough information to cope with the experience), confirmation of career choice (the students were asked how much this experience confirmed their career choice), and motivation (an open-ended question was posed in which respondents could express free thoughts).

### 2.4. Data Analysis

The interviews were analysed using the IPA hermeneutic–phenomenological method [22,23] which involves identifying meaning units, themes, and sub-themes consistent with the speaker’s language. The first step of the IPA involves immersing oneself in the original data. At this stage, it is important to read and re-read the transcripts, make margin notes, create a summary list of the margin notes, develop the emergent themes, and look for connections among them. Once each interview is individually analysed, the process is then extended to scan the entire set of transcripts for a full listing of sub-themes and themes; the transcripts are recoded, and the list of sub-themes and themes is finalized with quotations [22,23]. This methodology was suitable for the aim of this study to uncover the participants’ experience of care.

Three researchers began the analysis independently (SA, RB, AB), exploring one case after another. In case of disagreement, the three researchers went back to the original interview texts, attached notes, and reframed the shared themes under the supervision of the other researchers (NG, GV). A member of the research team (DFM) who had not participated in the interviews read the original uncoded transcripts to validate the findings.

### 2.5. Study Rigour

The authors applied this procedure with credibility, transferability and dependability to ensure rigour [19,24,25,26]. The accuracy and rigour of the data were ensured by researchers’ cross-checking: a first researcher (SA) dealt with the transcription of the data collected, and a second researcher (RB) listened to the audio recording while reading the transcription made by the first researcher to ensure that it accurately reflected the words of the interviewee. In order to ensure transferability, Table 1 shows the interview outline with IPA methodology and Table 2 shows the details of the participants.

## 3. Results

Thirteen students were included in the study: nine females and four males, with a mean age of 23.15 (range 21–28). All participants were not married and had no children. Data on sociodemographic characteristics and some closed-ended questions asked to the respondents are shown in Table 2.

Three themes and twelve subthemes emerged from the study (Table 3).

### 3.1. A Challenging Experience: It Is Not as Easy as It Seems

The students interviewed highlighted the difficulties related to having to face a totally new situation in a context as new as the COVID-19 vaccination centre that involved complex and standardized organization.

Other elements that made the experience complex and challenging concerned the difficulty in communicating with the team and integrating within it, feeling like the work was too much, and often feeling that the work completed was unrecognized. The negative influence of outside news and the lack of trust of some citizens also made the experience difficult.

#### 3.1.1. The Complexity of Organization and the Standardization of the Processes

Most students reported that they were particularly satisfied with the organization found within the COVID-19 vaccination centre, despite it being complex. One student, in addition to being happy on a personal level, reports hearing from friends and family who were particularly pleased with the centre’s organization.

Only one student reports complaints about the organization.

“In the first few days I felt like the stopgap, in the sense that they would put me where I was needed without a fixed position. [This], in my opinion, was also determined [by] a bad initial organization.” (No. 7).

Students reported a particular difficulty on days when there were multiple vaccines to administer. They describe those days as challenging: concentration had to be at a maximum, and the possibility of error was high. It emerges from the interviews that a standardization of processes often helped most students in performing their task in the best way possible.

“You have to have a mental scheme to stick to the order.” (No. 5).

#### 3.1.2. Lack of Communication Complicates Work

According to the interviewees, in some situations there was a lack of communication, which is a fundamental element for teamwork. Some students experienced miscommunications directly with the box team they belonged to, while others experienced miscommunication more broadly with other members of the organization.

“Difficulty [in] communicating with the physician, and with the colleague who was in the opposite stall.” (No. 5).

#### 3.1.3. Feeling out of Place

What emerges from the interviews of some students is a feeling of having felt “in the way”. Some, on the other hand, did not fully understand what their role was within the vaccination centre, feeling misunderstood and exploited as a “stopgap”.

“I felt a little bit like the stopgap in the sense that they would put me where I was needed without a set position.” (No. 7).

#### 3.1.4. My Work Was Not Recognized

Two students did not view the vaccination campaign positively; rather, they felt exploited.

“We were exploited. The shortage of trained nurses was filled by our support.” (No. 3).

Some students reported that their work was not recognized in a valid way, but only with the accumulation of internship hours that they would have preferred to spend in a hospital ward.

#### 3.1.5. The Negative Influence of News and Lack of Trust

The news of the discovery of vaccines has relieved the population of the feeling of helplessness in the face of an unknown pandemic. Some respondents felt that the media have generated psychological terrorism in the population, so much so that they have questioned the choice to vaccinate. Many of them felt anger, tension, and sadness regarding the lack of trust on the part of patients towards nursing and medical staff. The students interviewed felt anger at the time when patients preferred to refuse the dose of vaccine because it was different than the one they wanted to be vaccinated with.

“Another episode that stuck with me was [...] when the doctor made it clear [to a patient] that he should get the AstraZeneca vaccine [and] the patient started to freak out. Despite the doctor trying to convince him about the reliability of the vaccine and trying to reassure him [...] he signed up not to receive it.” (No. 9).

### 3.2. A Learning Experience Worth Living

For many interviewees, the experience at the COVID-19 vaccination centre was enriching and positive. This experience brought the students to understand a new reality, which many described as different from the hospital and ward reality. The experience, according to some interviewees, allowed them to consolidate and deepen relational and technical nursing skills. A common element in several of the interviews was the sense of gratitude that patients expressed towards the interviewees.

#### 3.2.1. The Added Value of an Unforeseen Experience

According to the interviewees, the experience allowed them to operate in a new context and in a particular situation, since it was not foreseen in their curriculum. Many were pleased and satisfied, as they were able to practice and refine their intramuscular administration skills and refine their interpersonal skills, personalizing their care.

“I had already participated in the COVID-19 vaccination campaign [...], and this gave me more confidence in the vaccine inoculation technique, but also from a relational point of view towards the patient. Being part of this campaign allowed me to experiment so much.” (No. 1).

One student interviewed said that the experience at the COVID-19 vaccination centre helped him affirm/validate/confirm his career choice.

#### 3.2.2. People’s Gratitude

During the interviews, students emphasized the importance of the gratitude they found from patients. For many of the interviewees, patients’ gratitude and emotion after the vaccination was gratifying.

“One child had a summer vacation assignment to describe his best day and had decided to talk about the day he got vaccinated.” (No. 6).

Some interviewees, during vaccinations, had the opportunity to relate to some patients who told and shared their stories about COVID-19.

#### 3.2.3. Sharing the Pandemic Experience

For many interviewees, the experience at the COVID-19 vaccination centre was also functional because of the synergy created and the sharing they had with all the health professionals present, and with the patients. One interviewee described the experience, to a degree, as alienating and monotonous.

“The first shift left me perplexed, [...] I spent the whole morning assembling syringes. It felt like alienating work to me [...], a monotonous activity that can be lightened by the people we interface with every day, the doctors and nurses at the station.” (No. 8).

#### 3.2.4. To Be up to the Task

Most of the students interviewed reported feeling tension and fear about making mistakes, especially when the risk of error was high, for example, when the number of vaccines to be administered was high.

“The work is challenging especially because on the days I participated we had four vaccines to administer so [...] the risk of error was very high.” (No. 5).

Despite this, one respondent reports experiencing positive fear, which allowed him to maintain a high level of concentration and perform the procedures correctly, while another student claims that his fears were gradually reduced by practicing.

“The fear was always there but it was a healthy fear. That right amount that allows you to work well.” (No. 10).

One student experienced moments of difficulty, not only because of the fear of making a mistake, but also because of the many activities he was completing at the same time as the collaborative experience in the vaccination campaign.

### 3.3. Teamwork and Trust leading to Professional Development

One of the most recurring positive aspects in the interviews was the collaboration perceived by the students during the vaccination campaign. Professionals made students participate in their work group, allowing them to feel part of the team and of a project that was fundamental for the entire community.

Despite the collaboration with nurses and physicians, some students felt that they had too much responsibility during the vaccination activity, often having to work independently, with the constant fear of making mistakes. However, some students consider the sense of responsibility experienced as an opportunity that could foster their growth and professional maturation.

#### 3.3.1. The Sense of Responsibility

Nearly all students reported feeling a sense of responsibility during the vaccination activity. Some respondents perceived the sense of responsibility as a factor contributing to growth and positive change for themselves, both as individuals and as nurses. Other students would have preferred more shadowing and support from a referring professional.

“We were supposed to be more and always supported by a nurse but that was not always the case, like in some cases when our tutor was busy with something else or was absent. We are therefore employed not as side-by-side students but rather as ‘done and done’ nurses.” (No. 3).

#### 3.3.2. Member of the Group

Students reported that they felt part of the team of nurses and physicians present at the COVID-19 vaccination centre, perceiving the involvement and collaboration with other professionals in a positive way. By becoming an integral part of the COVID-19 vaccination centre’s team of health professionals, students were able to feel like they were part of a project and recognize their usefulness to the entire community.

“I am glad I participated in this vaccination campaign because I made a contribution to the fight against COVID-19.” (No. 1).

The synergy that was created between the healthcare professionals and the students allowed the students to collaborate with future colleagues, grow professionally, and receive advice on future employment.

#### 3.3.3. Contributing to Change

A good portion of the students found their experience at the COVID-19 vaccination centre to be an opportunity to contribute to the fight against COVID-19, thus lending a hand to the community.

The thanks from colleagues, vaccinated individuals, and institutions further underscores the significance of the work performed by the nurses and students, thus increasing their pride in their role during the vaccination campaign.

“I am happy to give [my contribution] and when someone thanks me I say, «How nice» because I put a small piece of the puzzle of the picture that is [an] international [picture].” (No. 3).

The experience allowed the interviewees to have experience outside of the hospital setting in which internships are normally conducted, thus becoming an additional learning opportunity. Working with nurses in the COVID-19 vaccination centre allowed students a greater understanding of the importance of the role they will be taking on after graduation.

## 4. Discussion

The purpose of the study was to explore the lived experience of third-year nursing students during the COVID-19 vaccination campaign. Most studies in the literature that refer to nursing students investigate their adherence to COVID-19 vaccination, their knowledge about the vaccine, and their mental health during the pandemic [13,14,27,28,29]. No studies have been identified that explore the experiences of students who actively participated in the COVID-19 vaccination campaign.

### 4.1. The Organization

Being catapulted into a new and completely unfamiliar context created anxieties and concerns in the students, mainly related to the organizational complexity and the strong standardization of the processes, which made the students feel alienated and not able to understand their role within the structure. At times, students felt like a “stopgap” as they were moved from one part to another, perceiving a lack of organization and understanding from the team. According to most of the interviewees, the most complex days were those when there were multiple vaccines. These were stressful days where concentration had to be at a maximum. Communication plays a key role within teamwork and, according to some students, in this large project it sometimes failed, making it more complex. Recognition is another theme that was much discussed by the students, who felt that their work was not adequately recognized by some team members; on the contrary, some students felt that they were literally exploited by the system.

### 4.2. A New Experience

The experience at the COVID-19 vaccination centre allowed students to acquire not only new knowledge and technical skills but also social and communicative skills. The sense of gratitude expressed by the patients was very important to the students. It has been defined as an “enriching experience”, as it is not included in the study plan, thus allowing it to be an additional learning opportunity. According to Nania, the reorganization of learning activities and their habits could be a challenging question for students, requiring adequate institutional responses to enable them to manage anxiety and stress [30]. Carey’s study [31] also confirms the experience as a great learning opportunity for nursing students and educators. The students were able to consolidate the technical component of vaccine inoculation which, for some, had been poorly experienced in the ward. Working in a vaccination centre was a great learning opportunity not only for nursing students: medical students were involved at the same time in Griswold’s study [32]. An analysis of student evaluations revealed that the program taught fundamental clinical skills, increasing comfort giving intramuscular injections, and increasing comfort talking to patients about the COVID-19 vaccine [32].

Having an active part in the pandemic emergency generated a strong sense of responsibility and enthusiasm in the students. For some of them, it was further confirmation of their professional choice. As already stated by Nania, the mental health of college students can be significantly affected when dealing with public health emergencies such as the COVID-19 pandemic, and require careful attention and dedicated support from society, families and university leaders [30].

### 4.3. Working in a Team

Nursing students experienced an unavoidable feeling of not being up to the task entrusted to them, precisely because of their role as trainees less experienced than nurses. The greatest difficulties of the process stemmed from the anxiety and tension caused by the great responsibilities entrusted to students. Despite the difficulties encountered, students dwelled heavily on some positive aspects, first and foremost, teamwork. In most cases, nurses and doctors of the COVID-19 vaccination centre considered the students as real colleagues, making the students feel part of the team, thus giving them extra motivation to collaborate in the vaccination campaign. Joining the work team allowed students to get to know the professionals and learn from them.

As pointed out by Goni-Fuste et al., one of the factors that stimulates students’ willingness to collaborate is the moral duty to help future colleagues [12]. The theme of community was found also in Goode’s study [32]. Feeling like a member of the hospital/school and of nursing community of students and healthcare workers is an important theme; indeed, students crave safe, innovative clinical learning experiences, even amid a global pandemic, that help provide a sense of belonging [33,34,35,36].

Moreover, Carey’s study [31] also confirms the great learning opportunity not only for nursing students but also for educators. Indeed, the authors affirm that the vaccination experience gave educators a unique opportunity to observe interactions between students and vaccine recipients and to see how the students conducted patient education activities. This learning opportunity may be a good chance to re-think the way nursing students are educated. The lived experience of students may be useful in understanding how the organizations supporting them in their education can help them to undertake experiences and create welcoming contexts. The reported experience of feeling unrecognized by the organization is something that needs to be considered. In view of the changes taking place and the increased need for nursing staff, it is necessary for settings to work with the goal of creating effective and welcoming pathways to induction, including from an emotional perspective. Staff retention is increasingly becoming a topic of debate.

### 4.4. Strength and Limitations

This study has some strengths and limitations. The approach of the research opens up a scenario that is currently unexplored, such as the deepening of the vaccination campaign experience. However, because our findings are based on the interviews of thirteen students talking about their own experiences, results should be used with caution. It would have been interesting to compare the experiences of these students with those of other universities’ students. Despite efforts to recruit a homogenous sample, students differed in aspects such as gender, which could be one factor influencing the reported experience [37,38]. Although debriefing sessions on the experience have been conducted, it might be useful to use this work to capture the students’ lived experience at a distance and to create more reflection on the experience.

Given that they were chosen based on their availability, it is possible that those who participated were more willing to share their experience. Despite these limitations, it is clear from the resulting themes that there were commonalities across narratives, providing valuable insights into this experience.

## 5. Conclusions

This study highlighted the lived experience of a group of Italian third-year nursing students during the COVID-19 vaccination campaign using the IPA methodology. This is an issue that has been little addressed in the present literature, but one that may have important implications, especially with regard to the education of nursing students.

The main themes that emerged relate to organization, new experiences, and teamwork. With regard to organization, students had difficulty understanding organization and, at times, felt overwhelmed. An important aspect is the high level of responsibility of the students, an element that for some of them was a cause of anxiety and fear, while for others it was an opportunity for professional growth. This study allowed us to investigate the relationship that is created between students and nurses in the workplace. It is evident from the interviews that, in most cases, teamwork was a positive factor in the experience, although in some cases students felt overwhelmed. The experience that was proposed to the students represented a novelty for the degree program. Students emphasized, above all, the positive aspects of having had the opportunity to participate in the vaccination campaign, contributing to helping the entire community in the fight against COVID-19.

The results of the present study will help educators understand the importance of providing information to students before they take part in vaccination campaigns. This factor was one of the negative aspects most highlighted by the students who, in their interviews, stated quite frequently that having more information would have helped them to face the experience with more confidence and readiness. The fact that the students interviewed welcomed the experience proposed to them, despite the fact that it was new, may be a useful cue for degree planning, which could more frequently propose alternative opportunities to the internship experience in a hospital setting.

## Figures and Tables

**Table 1 healthcare-10-01985-t001:** Interview outline with IPA methodology.

Objective	Open-Ended Questions
Exploring the experience of third-year nursing students during the COVID-19 vaccination campaign.	How would you describe your experience of joining the vaccination campaign? How did you feel about a new experience that was different from previous ones? Did you experience moments of difficulty? What allowed you to persevere in the activity despite the difficulties? We ask you to recount an episode that made you think. What happened? What were you thinking while it was happening? Do you think this experience influenced your achievement of professional skills? If so, what skills did it influence and how?

**Table 2 healthcare-10-01985-t002:** Sociodemographic characteristics.

		N
Gender	Male	4
	Female	9
Age	Mean (range)	23.15 (21–28)
COVID-19 experience	Yes	5
	No	8
Education: Do you feel your preparation was sufficient to handle the experience?	Yes	12
	No	1
Information: Do you feel you received enough information to deal with the experience?	Yes	11
	No	2
Career choice: How much did this experience validate your career choice? 0 (a little)–10 (a lot)	Mean (range)	7.85 (4–10)

**Table 3 healthcare-10-01985-t003:** Themes and subthemes.

Themes	Subthemes
A challenging experience: It is not as easy as it seems	The complexity of the organization and the standardization of processes Lack of communication complicates work Feeling out of place My work was not recognized The negative influence of news and lack of trust
A learning experience worth living	6. The added value of an unforeseen experience 7. People’s gratitude 8. Sharing the pandemic experience 9. To be up to the task
Teamwork and trust leading to professional development	10. The sense of responsibility 11. Member of the group 12. Contributing to change

## Data Availability

Not applicable.

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
