# Peer review of "Frontline Involvement in Population COVID-19 Vaccinations: Lived Experience of Nursing Students"

_healthcare, 2022, doi:10.3390/healthcare10101985_

Round 1

Reviewer 1 Report

This article explored the experiences of nursing students explored the lived experience of third-year nursing students during the COVID-19 vaccination campaign in Italy. This is an important student that can contribute to the role that nursing students played during the COVID-19 pandemic, as well as the challenges faced. However, there are several challenges with this paper and I will highlight these below:

1. There is a need for major language editing. For example, the first sentence in the abstract (lines 12-13, page 1) and the sentence that started line 24, page 1 does not make sense. There are many of 

2. Methodology: There are many issues needing attention in the methodology section. For example, why were third-year students selected for this study? Are these students in their final year? How many students worked in the Vaccination centre before 13 were selected? Did the students take part in the vaccination campaign as part of their student placement or were they volunteers? 

3. Participants: How did you decide the participants who were the 'most informative'? What is IPA? - The acronym was used but not spelt out

4. Data collection: What do 'very open-ended questions' mean? 

5. Phenomenon of interest: While the study claims to be interested in lived experience, it is not clear what the phenomenon or the experience being explored is. In some sections, it was written that aim was the experience lived by third-year students, in other parts, it was said the study was exploring their perceptions during the vaccination campaign. Again, what exactly is the experience that this study is trying to understand?

6. Please clarify what 'remote modes' are.

7. Data analysis: What steps were taken to analyse the data and what approach was used? Page 3, line 103, kindly clarify 'themes and themes' and on the same page (line 105), is 'caring' the phenomenon?

8. Results: The data need to be re-examined because all the themes did not capture the subthemes and some subthemes are not even supported with direct quotes. It was not clear how themes were derived. For example, there was no quote for 3.1.4 and 3.1.1, 3.1.2 and 3.2.4 are more like challenges, while 3.2 and 3.2.1 are about the benefits of working at the vaccination centre. 3.2.2 is about strategies for coping with challenges, while 3.3.1 is unrelated to the theme of teamwork.

Discussion: The discussion was a summary of findings, and there is no intellectual curiosity in it.

Author Response

This article explored the experiences of nursing students explored the lived experience of third-year nursing students during the COVID-19 vaccination campaign in Italy. This is an important student that can contribute to the role that nursing students played during the COVID-19 pandemic, as well as the challenges faced. However, there are several challenges with this paper and I will highlight these below:

Dear thank you for your time, support and suggestions.

  1. There is a need for major language editing. For example, the first sentence in the abstract (lines 12-13, page 1) and the sentence that started line 24, page 1 does not make sense. There are many of 

Answer: Thank you for the suggestion. We tried to revise the whole article and adjust, hopefully it has improved.

  1. Methodology: There are many issues needing attention in the methodology section. For example, why were third-year students selected for this study? Are these students in their final year?

Answer: Thank you for the suggestion. We clarified in the methods.

  1. How many students worked in the Vaccination centre before 13 were selected? Did the students take part in the vaccination campaign as part of their student placement or were they volunteers? 

Answer: Thank you for the suggestion. We clarified in the methods.

  1. Participants: How did you decide the participants who were the 'most informative'? What is IPA? - The acronym was used but not spelt out

Answer: Thank you for the suggestion. We clarified in the methods.

  1. Data collection: What do 'very open-ended questions' mean? 

Answer: Thank you for the suggestion. The term is confusive, we tried to clarified.

  1. Phenomenon of interest: While the study claims to be interested in lived experience, it is not clear what the phenomenon or the experience being explored is. In some sections, it was written that aim was the experience lived by third-year students, in other parts, it was said the study was exploring their perceptions during the vaccination campaign. Again, what exactly is the experience that this study is trying to understand?

Answer: Thank you for the suggestion. The phenomenon of interest was the experience lived by third-year students during the COVID-19 vaccination campaign. We clarified in the article as suggested.

  1. Please clarify what 'remote modes' are.

Answer: Thank you for the suggestion. We clarified in the methods. We used the term videocall in online platform in order to described better the method used.

  1. Data analysis: What steps were taken to analyse the data and what approach was used?

Answer: In the first paragraph of the data analysis we reported the steps that the IPA methodology recommends for data analysis, trying to describe it better. we hope it will fit and answer the suggestion

  1. Page 3, line 103, kindly clarify 'themes and themes' and on the same page (line 105)

Answer: Thank you for the suggestion. We modified in themes and sub-themes

  1. Results: The data need to be re-examined because all the themes did not capture the subthemes and some subthemes are not even supported with direct quotes. It was not clear how themes were derived. For example, there was no quote for 3.1.4, 1.1, 3.1.2 and 3.2.4 are more like challenges, while 3.2 and 3.2.1 are about the benefits of working at the vaccination centre. 3.2.2 is about strategies for coping with challenges, while 3.3.1 is unrelated to the theme of teamwork.

Answer: Thank you for the suggestion. We revised with other author the data and we modified. Subthemes 3.2.4 (The negative influence of news and lack of trust) and 3.3.1 (Feeling up to the task) were moved on Themes 1: Learning facing unexpected and not facilitative situations: It is not as easy as it seems. We added the quotations.

  1. Discussion: The discussion was a summary of findings, and there is no intellectual curiosity in it.

Answer: Thank you for the suggestion. We added some reflections.

Reviewer 2 Report

The study entitled "Frontline involvement in population COVID-19 vaccinations: 2 Lived experience of nursing students" presents an interesting insight into the students who had to be called upon to strengthen health systems during the pandemic. It is a well-structured and well-written study that deserves to be published.  Here are some suggestions for improvement: 

- Summary: Recommending removing this sentence from the Conclusions: "This study analysed the experiences of nursing students who participated in the COVID-19 vaccination campaign.

- Line 45: Correctly identify the acronym PPE.

- Line 73: Correctly identify the acronym IPA.

- Revise line 71 that talks about patients instead of students.

- The instrument, did an expert panel pass? The questions are very appropriate and well posed.

- Line 295, you say: “The experience at the COVID-19 vaccination center has allowed the students to acquire new knowledge and technical skills”. But you should also add that they have acquired social and communicative skills.

Author Response

The study entitled "Frontline involvement in population COVID-19 vaccinations: 2 Lived experience of nursing students" presents an interesting insight into the students who had to be called upon to strengthen health systems during the pandemic. It is a well-structured and well-written study that deserves to be published.  Here are some suggestions for improvement: 

 Dear thank you for your support and suggestions.

- Summary: Recommending removing this sentence from the Conclusions: "This study analysed the experiences of nursing students who participated in the COVID-19 vaccination campaign.

Answer: done! Thank you for the suggestion.

- Line 45: Correctly identify the acronym PPE.

Answer: done! Thank you for the suggestion.

- Line 73: Correctly identify the acronym IPA.

Answer: done! Thank you for the suggestion.

- Revise line 71 that talks about patients instead of students.

Answer: done! Thank you for the suggestion.

- The instrument, did an expert panel pass? The questions are very appropriate and well posed.

Answer: done! Thank you for the suggestion.

- Line 295, you say: “The experience at the COVID-19 vaccination center has allowed the students to acquire new knowledge and technical skills”. But you should also add that they have acquired social and communicative skills.

Answer: done! Thank you for the suggestion.

Reviewer 3 Report

To clarify this research report, I have the following recommendations:

Line 52-54 (delete as redundant)

Line 91 delete patients, insert participants

Line 96 delete investigated, insert asked

Line 103: ...units, themes and sub-themes (add sub-)

Line 114: delete was, insert were

Table 1: ....explaining the experience experienced by Nursing students:  change to:  explaining the experience of Nursing students

Table 3:  don't center align subthemes - it makes it difficult to read

Line 285 ...stop hole - change to stop gap

Line 292...here you say that the students felt their work was not adequately recognized; however in the data analysis, it was mentioned several times that the students were pleased with all the gratitude shown by patients. This discrepancy needs to be cleared up - was it only the team members who didn't recognize students' work?  And it should be mentioned in the discussion that thank yous from patients were important to students as this came through in the data analysis section as important. 

Line 322-23: this is not a complete sentence; delete the word "that" after research and this will make it a complete sentence

Line 341: ...felt like too much... change to:  felt overwhelmed

Line 346 ...felt like too much....change to felt overwhelmed

Finally, it would seem to me important to suggest something about debriefing students after an experience like this...perhaps the research itself constituted a debriefing, but hopefully the faculty also did debriefing with the students.

Author Response

To clarify this research report, I have the following recommendations:

Dear reviewer thank you for your support and suggestion.

Line 52-54 (delete as redundant)

Answer: done! Thank you for the suggestion.

Line 91 delete patients, insert participants

Answer: done! Thank you for the suggestion.

Line 96 delete investigated, insert asked

Answer: done! Thank you for the suggestion.

Line 103: ...units, themes and sub-themes (add sub-)

Answer: done! Thank you for the suggestion.

Line 114: delete was, insert were

Answer: done! Thank you for the suggestion.

Table 1: ....explaining the experience experienced by Nursing students:  change to:  explaining the experience of Nursing students

Answer: done! Thank you for the suggestion.

Table 3:  don't center align subthemes - it makes it difficult to read

Answer: done! Thank you for the suggestion.

Line 285 ...stop hole - change to stop gap

Answer: done! Thank you for the suggestion.

Line 292...here you say that the students felt their work was not adequately recognized; however in the data analysis, it was mentioned several times that the students were pleased with all the gratitude shown by patients. This discrepancy needs to be cleared up - was it only the team members who didn't recognize students' work?  And it should be mentioned in the discussion that thank yous from patients were important to students as this came through in the data analysis section as important. 

Answer: Thanks for the comment. We clarified the discrepances you mention. we specify in the article that are some team members who didn't recognize students' work. We added the gratitude from patients in the discussion

Line 322-23: this is not a complete sentence; delete the word "that" after research and this will make it a complete sentence

Answer: done! Thank you for the suggestion.

Line 341: ...felt like too much... change to:  felt overwhelmed

Answer: done! Thank you for the suggestion.

Line 346 ...felt like too much....change to felt overwhelmed

Answer: done! Thank you for the suggestion.

Finally, it would seem to me important to suggest something about debriefing students after an experience like this...perhaps the research itself constituted a debriefing, but hopefully the faculty also did debriefing with the students.

Answer: Thank you for the suggestion. We added in the section strength and limitations

Round 2

Reviewer 1 Report

Thank you for the opportunity to re-review this manuscript. I want to commend the authors for the amendments done. There is improvement in the paper. However, there are still some areas needing further revision.

1. The grammar still needs further improvement. For example, in the abstract's conclusion, it was stated that "the participation in the vaccination campaign was a novelty for students in their degree program. Nevertheless ...". In English, nevertheless is a synonym of however and when used, you expect a contrast of the previous sentence. The first sentence said the experience of vaccination was new for the students. The second sentence then said 'it was a positive experience. For nevertheless to be an appropriate use of words, the first sentence should talk about the challenges or negatives of students as vaccinators. 

2. Under 2.2 Setting and participation, remove the comma between 'selected' and 'by' on line 72. Also, give the final sample size from the purposive sampling in this section. Also when you said, 'just finished their experience', is there any timeframe? 

3. Under 2.3 data collection, replace 'built' (line 85) with developed by the research team. Likewise, remove 'female' on page 87 as the gender of the researcher here is not relevant. You can add the initials of the particular researcher who collected the data.

4. Under 2.4 analysis, clarify 'adherence on line 107 and remove 'into' on line 116. 

5. Under study rigour, revise the first sentence (line 125) to: 'the authors have applied the procedure of credibility, transferability and dependability to ensure rigour. Also, address the consistency of the spelling of rigour. The steps to ensure transferability were not addressed.

6. The themes still need to be revised further, and the presentation needs to be more focused. In theme 1, it appears the main message was difficulties/challenges experienced. Revised the name as such and at the end of the theme description, list standardisation/organisational complexity, communication, feeling in the way (more like feeling out of place), feelings of under-appreciation, and public perception as the factors that led to difficulties for the students. In each of these sub-themes, direct quotes should then be presented to show the range of perceptions. Feeling up to the tasks subtheme is confusing because all the other themes had a negative connotation. Lumping all these together is going to affect the message that is being across. As the second theme 'a learning worth living' has a more positive spin, the last subtheme in theme one should be revised and moved to theme 2. 

7. Under 3.2.1, clarify or revise the use of the word 'confirming' on line 225 (page 5). 

8. Under 3.2.2 (line 230), the quote does not make meaning. Kindly find a more appropriate and illustrative quote. Otherwise, the subtheme should be removed. It is important not to force all the findings into one paper.

9. Again, the discussion is yet to be improved because it is a discussion of some implications and a rehearsing of the findings. I have listed some articles on student nurses' roles as vaccinators that can be used to compare and contrast what was found in this study. Also, the discussion can be more impactful by looking at the experiences of other medical and pharmacy students who worked as vaccinators. This will help with the application, transferability and overall quality of this paper.

A. Unexpected Learning Opportunities for Nursing Students (DOI: 10.1097/01.NAJ.0000815428.23462.37).

B. Nursing students' reflections on vaccine administration during the COVID-19 global pandemic (https://doi.org/10.1016/j.profnurs.2022.08.005)

C. If the public can vaccinate, why not students? Review of a student nurse placement in a mass vaccination centre (https://www.magonlinelibrary.com/doi/abs/10.12968/bjon.2022.31.7.386).

There are also many studies on other students such as pharmacy, medical and so on who also worked as vaccinators. Comparing their experiences with the current study will improve the discussion section and elevate the overall quality of this paper.

Author Response

Dear reviewer

We performed our best in order to modified and increased the quality of the article following your suggestion

  1. The grammar still needs further improvement. For example, in the abstract's conclusion, it was stated that "the participation in the vaccination campaign was a novelty for students in their degree program. Nevertheless ...". In English, nevertheless is a synonym of however and when used, you expect a contrast of the previous sentence. The first sentence said the experience of vaccination was new for the students. The second sentence then said 'it was a positive experience. For nevertheless to be an appropriate use of words, the first sentence should talk about the challenges or negatives of students as vaccinators. 

OK – done. A British native speaker read the paper and made some changes, hopefully they will be enough

  1. Under 2.2 Setting and participation, remove the comma between 'selected' and 'by' on line 72. Also, give the final sample size from the purposive sampling in this section. Also when you said, 'just finished their experience', is there any timeframe? 

OK - done

  1. Under 2.3 data collection, replace 'built' (line 85) with developed by the research team. Likewise, remove 'female' on page 87 as the gender of the researcher here is not relevant. You can add the initials of the particular researcher who collected the data.

OK - done

  1. Under 2.4 analysis, clarify 'adherence on line 107 and remove 'into' on line 116.

OK – done, we propose the term CONSISTENT

  1. Under study rigour, revise the first sentence (line 125) to: 'the authors have applied the procedure of credibility, transferability and dependability to ensure rigour. Also, address the consistency of the spelling of rigour. The steps to ensure transferability were not addressed.

OK – done

  1. The themes still need to be revised further, and the presentation needs to be more focused. In theme 1, it appears the main message was difficulties/challenges experienced. Revised the name as such and at the end of the theme description, list standardisation/organisational complexity, communication, feeling in the way (more like feeling out of place), feelings of under-appreciation, and public perception as the factors that led to difficulties for the students. In each of these sub-themes, direct quotes should then be presented to show the range of perceptions. Feeling up to the tasks subtheme is confusing because all the other themes had a negative connotation. Lumping all these together is going to affect the message that is being across. As the second theme 'a learning worth living' has a more positive spin, the last subtheme in theme one should be revised and moved to theme 2. 

OK – done. We revised and modified as you suggested.The sub-theme “Feeling up to the task” was modified in “To be up to the task” and moved from theme 1 to theme 2

  1. Under 3.2.1, clarify or revise the use of the word 'confirming' on line 225 (page 5). 

OK – done, we propose the term affirm/validate/confirm

  1. Under 3.2.2 (line 230), the quote does not make meaning. Kindly find a more appropriate and illustrative quote. Otherwise, the subtheme should be removed. It is important not to force all the findings into one paper.

OK – done

  1. Again, the discussion is yet to be improved because it is a discussion of some implications and a rehearsing of the findings. I have listed some articles on student nurses' roles as vaccinators that can be used to compare and contrast what was found in this study. Also, the discussion can be more impactful by looking at the experiences of other medical and pharmacy students who worked as vaccinators. This will help with the application, transferability and overall quality of this paper.
  2. Unexpected Learning Opportunities for Nursing Students (DOI: 10.1097/01.NAJ.0000815428.23462.37).
  3. Nursing students' reflections on vaccine administration during the COVID-19 global pandemic (https://doi.org/10.1016/j.profnurs.2022.08.005)
  4. If the public can vaccinate, why not students? Review of a student nurse placement in a mass vaccination centre (https://www.magonlinelibrary.com/doi/abs/10.12968/bjon.2022.31.7.386).

OK – done. For the paper C: We've tried searching but can't access

There are also many studies on other students such as pharmacy, medical and so on who also worked as vaccinators. Comparing their experiences with the current study will improve the discussion section and elevate the overall quality of this paper.

OK – done

Thank you for your supervision.

Kind Regards

The authors